# BATCHPROMPT: ACCOMPLISH MORE WITH LESS

**Jianzhe Lin, Maurice Diesendruck, Liang Du, Robin Abraham**
Microsoft

## ABSTRACT

The ever-increasing token limits of large language models (LLMs) have enabled long context as input. Many LLMs are trained and fine-tuned to perform zero/few-shot inference using instruction-based prompts. Prompts typically include a detailed task instruction, several examples, and a single data point for inference. This baseline is referred to as "SinglePrompt" in this paper. In terms of token count, when the data input is small compared to instructions and examples, this results in lower token utilization, compared with encoder-based models like fine-tuned BERT. This cost inefficiency, affecting inference speed and compute budget, counteracts many of the benefits that LLMs offer. This paper aims to alleviate this problem by batching multiple data points in each prompt, a strategy we refer to as "BatchPrompt". We improve token utilization by increasing the "density" of data points, however, this cannot be done naively. Simple batching can degrade performance, especially as batch size increases, and data points can yield different answers depending on their position within a prompt. To address the quality issue while retaining high token utilization, we introduce Batch Permutation and Ensembling (BPE) for BatchPrompt – a simple majority vote over repeated permutations of data, that recovers label quality at the cost of more token usage. To counterbalance this cost, we further propose Self-reflection-guided EArly Stopping (SEAS), which can terminate the voting process early for data points that the LLM handles confidently. Our comprehensive experimental evaluation demonstrates that BPE + SEAS can boost the performance of BatchPrompt by a striking margin on a range of popular NLP tasks, including question answering (Boolq), textual entailment (RTE), and duplicate questions identification (QQP). This performance is even competitive with/higher than single-data prompting (SinglePrompt), while using far fewer LLM calls and input tokens. At batch size 32, our BatchPrompt + BPE + SEAS uses 15.7% the number of LLM calls, and achieves: **Boolq** accuracy 90.6% $\rightarrow$ 90.9% with 27.4% tokens, **QQP** accuracy 87.2% $\rightarrow$ 88.4% with 18.6% tokens, **RTE** accuracy 91.5% $\rightarrow$ 91.1% with 30.8% tokens. We hope our simple yet effective approach will shed light on the future research of large language models. Code: github.com/microsoft/BatchPrompt

## 1 INTRODUCTION

A recent trend in the NLP landscape is adapting large language models in various practical applications, including conversational interface, question answering, and context summarization. These downstream tasks are primarily performed through prompting: The task specification and data are combined as an input context to the language model, which generates a completed text to be returned. The length of this input ranges from hundreds to thousands of tokens. Recent progress in hardware and algorithms has enabled longer context windows with longer inputs. This trend forces us to reflect on whether including only a single data sample as input is an efficient setting for prompting. Instead, a better input might contain the task specification (for simplification, task specification represents both task description and examples of demonstrations in the paper) with batched data.

However, dealing with long context input is not easy for large language models which are implemented with Transformers. Transformers scale poorly to long sequences, which leads to a severe performance decrease in language models Liu et al. (2023a). The reason might be that the complexity of the self-attention module in a transformer is quadratic with the input length. As prompting with batched data inevitably stretches input length which decreases performance, finding a better batch prompting strategy instead of naively batching data is necessary.

Through experiments, we observe that LLM performance varies significantly when data is in different positions and orders, represented in Fig. 1 with ascending data index. This change of performance could be due to the autoregressive nature of the LLM decoder, which predicts each output token conditioned on previous outputs. This means that each answer is generated with a different context, with different token distances to the original task specification and data.

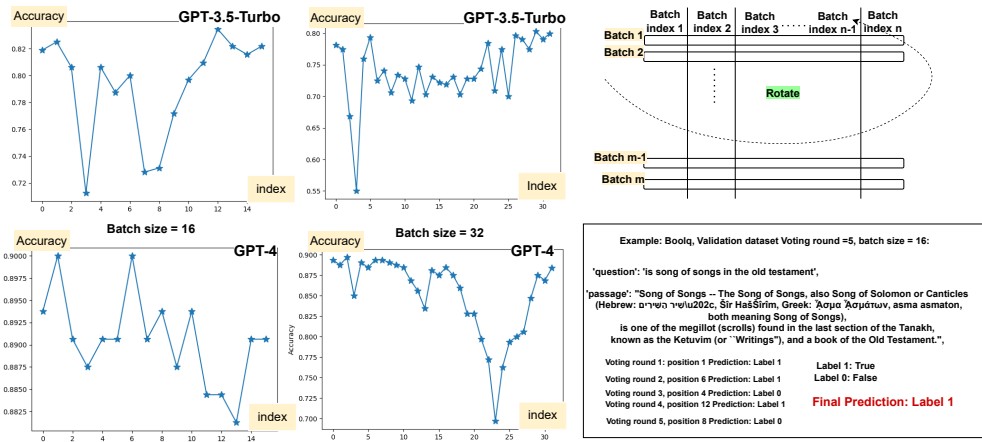

Figure 1: An example of output from gpt-3.5-turbo/gpt-4 with prompts containing data in different batch sizes (16, 32). We use the results of Boolq dataset as an example. We have $m \times n$ data in total with $m$ batches of size $n$, represented with batch indices 1 to $n$. We rotate the batches $n$ times to ensure that each data have visited every batch index. Finally, we calculate the average accuracy for each position, specifically, an average accuracy of all $m \times n$ samples at each position. The prediction varies largely. Also, with the increase of batch size, the overall accuracy decreases.

Based on this observation, we propose Batch Permutation and Ensembling (BPE) to boost performance of BatchPrompt, which leverages the intuition that a uniformed LLM output for multiple batches, with the same data assembled in diverse orders, will be more promising. Specifically, instead of sampling the data sequence in its original order, we permute the data in each batch. Different orders induce different outputs, and the ensemble is achieved through majority voting.

Compared with prior methods that annotate data and train additional models, BPE is far simpler, and works off-the-shelf with pre-trained LLMs, requiring no extra human annotation. Also, different from ensemble learning, BPE can be viewed as "self-ensembling" that works on top of a single language model. An example of the BPE process can be found in Fig. 2.

The final goal of BatchPrompt is to accomplish more data processing with fewer tokens/LLMs calls. We find that the number of LLM calls decreases substantially, while the decrease of total tokens depends on the proportion of task tokens to data tokens, as well as the number of voting rounds. Generally, we find there is an obvious decrease in tokens used when the number of voting rounds in BPE is less than 10, while the batch size is higher than 64. Larger batch size with fewer voting rounds will boost the frugality. In terms of performance, the accuracy of BatchPrompt can even be competitive with single-data prompting when the number of voting rounds is larger than five.

To further boost frugality even with many voting rounds, we propose a Self-reflection-guided EArly Stopping (SEAS) method. By prompting LLMs to provide a confidence label in addition to a prediction, we encourage shorter voting rounds where possible. For each specific data, if consecutive "confident" ratings are returned, it could be redundant to continue voting on that data, and prediction can stop early. In this way, around 80% of data can be answered within two voting rounds, and the total tokens used can be kept low. A side benefit of such self-reflection is a boost in prediction accuracy, which is also demonstrated in experiments.

We thoroughly demonstrate the effectiveness of BatchPrompt with BPE and SEAS on three datasets with different tasks from Glue and Super Glue benchmark (Boolq, QQP, RTE). Also, we extend the BatchPrompt to a recently popular arithmetic reasoning task (GSM8K) in a small side experiment.

A concrete example of the effectiveness of the proposed method is below, where for batch size 32 and five voting rounds, the total tokens used for Boolq, QQP, and RTE decrease by 72.6%, 81.4%, and 69.2%, respectively. Note that tokens here are all input tokens - we do not count LLM-generated output tokens which cannot be controlled/decreased. LLMs calls decrease by 84.4%, 90.6%, and 84.4%, respectively; and accuracies are competitive with SinglePrompt (single-data prompting), $90.6\% \rightarrow 90.9\%$, $87.2\% \rightarrow 88.4\%$, and $91.5\% \rightarrow 91.1\%$, respectively, as in Fig. 4.

To summarize, our contributions are the following:

- We propose BatchPrompt, an efficient prompting technique for LLMs;
- We propose the BPE method to boost the performance of BatchPrompt;
- We propose the SEAS method to boost both the accuracy and efficiency of BatchPrompt, and keep LLM tokens/calls/cost low;
- We show that BatchPrompt with BPE and SEAS achieves promising performance on benchmark tasks.

## 2 BATCHPROMPT

A typical prompt includes a task specification and data to be processed or labeled. It is less efficient to process data one-by-one, and more efficient to use batching. However, we find that when prompt lengths increase, the performance decreases correspondingly. To overcome this limitation, we propose the Batch Permutation and Ensembling (BPE) method, which repeats multiple voting rounds for each batch, each time using data in a different order. It is natural to suppose that LLMs can process batched data in diverse orders, and we induce this diversity via permutation. Although the model might still make mistakes for some data at specific positions in specific voting rounds, such wrong answers are less likely to be the same. However, the correct answer generated for a specific data located at different positions in different voting rounds, tends to have greater agreement compared to the incorrect one. BPE is compatible with many voting strategies.

Thinking conceptually, when people prioritize and ascribe different attention to different information, each piece of information has a unique effect on the other. The same may be true for LLMs processing data in batches. We formulate the problem as follows. Suppose the current batch of data with batch size $N$ is $D = \{d_1, d_2, ..., d_N\}$, the answers to each data are $A = \{a_1, a_2, ..., a_N\}$, and the orders in $K$ permutations are $S = \{s_1, s_2, ..., s_K\}$. We formulate the majority voting as:

$$\arg\max_a \sum_{k=1}^K \mathbb{1}(a_n^k = a), \forall n \in \{0, 1, ..., N\} \tag{1}$$

to generate the final answer for $d_n$. However, in different permutations, the answer $a_n$ to the data $d_n$ is not only conditioned on the task specification, but also on $\{a_1, a_2, ...a_{n-1}\}$, which can be viewed as the context of $a_n$. Therefore, we can further formulate the majority voting as:

$$\arg\max_a \sum_{k=1}^K \mathbb{1}(a_n^k = a \mid prompt, a_1^k, a_2^k, ..., a_{n-1}^k), \forall n \in \{0, 1, ..., N\} \tag{2}$$

We can find from this equation that the output answer $a_n$ for data $d_n$ will differ with different permutations due to the change of context. This observation can explain why BatchPrompt with BPE can perform even better than SinglePrompt.

We further explore a weighted majority voting method, in which the weight is generated by the LLM itself to avoid extra training. We call this process self-weighted majority voting (sw-mv), and provide an example in Fig. 2. Inspired by the reflexion idea Shinn et al. (2023), in the task specification, we add this text: *If you are confident in your output class, append a "confident" at the end of the label; else, append a "not confident".* This causes the LLM to automatically generate a binary weight. If the generated answer is "confident" we assign weight as $w = 1$; if "not confident", $w = \alpha, \alpha \in [0, 1]$. In experiments, we empirically set $\alpha$ to 0.2. We can formulate this self-weighted majority voting as:

$$\arg\max_a \sum_{k=1}^K w_n \cdot \mathbb{1}(a_n^k = a \mid prompt, a_1^k, a_2^k, ..., a_{n-1}^k), \forall n \in \{0, 1, ..., N\} \tag{3}$$

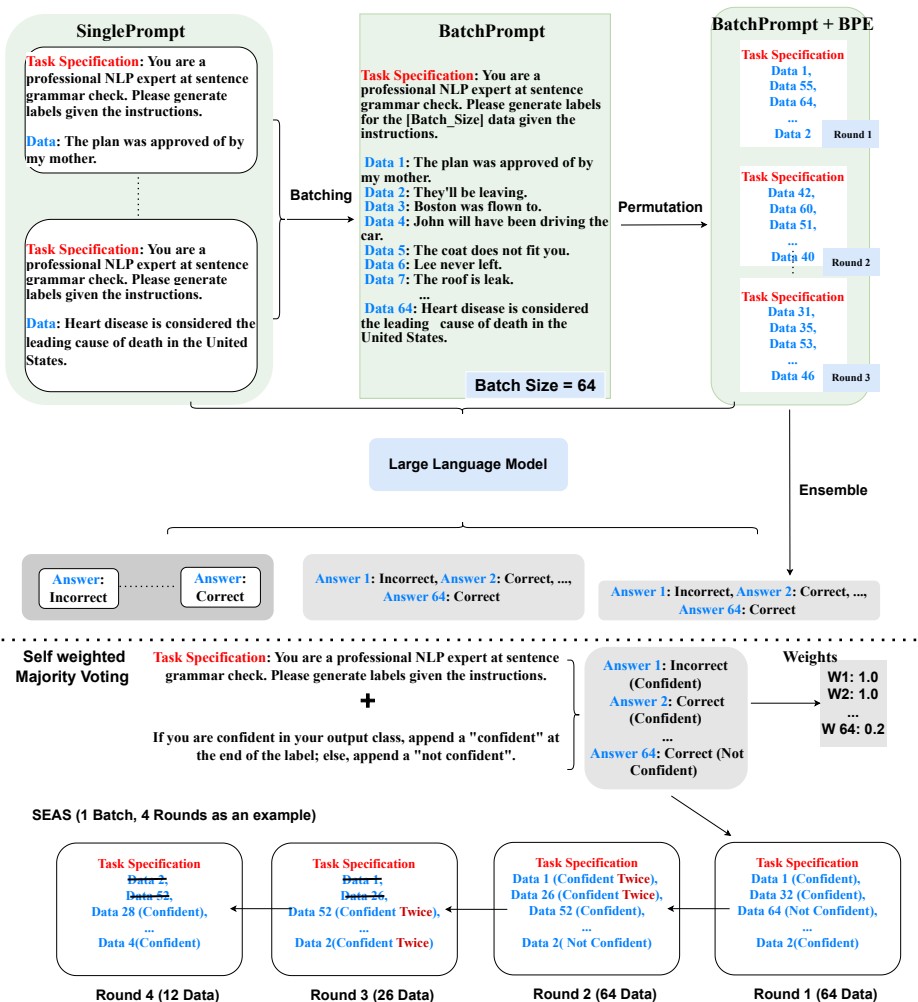

Figure 2: The flowchart for the proposed BatchPrompt. In the top row, the left column is SinglePrompt, calling once for each data sample; the middle column shows BatchPrompt, which naively batches data; and the right column is BatchPrompt + BPE, which adds batch permutation. The middle row shows corresponding model outputs, noting the Ensemble method on the right, for deriving answers with BPE. The third and last rows show the proposed self-weighted majority voting and SEAS. By adding confidence ratings to the task specification, weights can be self-generated by LLMs. Through SEAS, the token number can be minimized while the accuracy can be improved.

We envision more, and more sophisticated, voting strategies, but focus in this work on regular majority voting (mv) and self-weighted majority voting (sw-mv).

## 3 SEAS

A deeper thinking of the proposed self-weighted majority voting (sw-mv) enables an even more aggressive strategy for efficiency that we call Self-reflection-guided EArly Stopping (SEAS). When a batch contains easy data samples, instead of continuing to vote many times, we utilize repeated "confident" labels to truncate the procedure for those samples. As a result, the effective batch size is reduced and total number of voting rounds can be shorter than the maximum, all while maintaining high accuracy. The method is described using the pseudo-code in Alg. 1.

The SEAS method not only effectively cuts down the token count, but can also boost accuracy. As voting rounds continue, easy samples will be removed earlier, leaving fewer/harder samples for later

rounds. The harder samples might also become easier to predict, due to smaller effective batch size in later rounds. For example, at the beginning in voting round 1, the batch size is 32. Later in voting round 5, there might be only 2 hard samples without consistent "confident" predictions left in the batch. The LLM now only needs to predict labels/answers of these 2 samples, which might be more accurate compared to a prediction with 32 batched samples. One alternative is to fill in each batch to the full batch size as easy samples are finished, but this sacrifices the side benefit of predicting hard samples with smaller batches, and is not selected in SEAS.

---

**Algorithm 1** SEAS

---

**Function** *SEAS(batch D, batch size N, permutation S(), voting round K, LLM)*

Initialize $last\_answer[N] \leftarrow None$

Initialize $last\_confidence[N] \leftarrow None$

Initialize $results[N] \leftarrow EmptyDictionaries$

Initialize $active\_indices \leftarrow \{1, 2, \ldots, N\}$

**for** $k \leftarrow 1$ *to* $K$ **do**

    $D' \leftarrow S(D)$

    $answers, confidences \leftarrow \text{LLM}(D')$

    **for** $i \in active\_indices$ **do**

        $answer \leftarrow answers[i]$

        $confidence \leftarrow confidences[i]$

        $results[i][answer] + = 1$

        **if** $k > 1$ **and** $confidence ==$ *"confident"* **and** $last\_confidence[i] ==$ *"confident"*

        **and** $answer == last\_answer[i]$ **then**

            $active\_indices \leftarrow active\_indices \setminus \{i\}$

        $last\_answer[i] \leftarrow answer$

    **end**

**end**

**for** $i \leftarrow 1$ *to* $N$ **do**

    $final\_answer[i] \leftarrow MajorityVoteOfResults[i]$

**end**

---

## 4 EXPERIMENTS

### 4.1 EXPERIMENTAL SETTINGS

**Tasks and datasets:** We conduct our experiments on several NLP tasks, including question answering (Boolq), duplicate text identification (QQP), and textual entailment (RTE).

**Boolq:** Boolean Questions (Boolq) is a question-answering dataset for yes/no questions containing 15942 examples (9427 for training, 3270 for validation, 3245 for testing). Each example contains a triplet of (question, passage, and answer). Question and passage are used as input prompts and the answer is to be generated.

**QQP:** Quora Question Pairs (QQP) dataset Wang et al. (2017) consists of >400,000 question pairs, and each question pair is annotated with a binary value indicating whether the two questions are paraphrases of each other. Question pairs are used as input prompts while the label is to be generated.

**RTE:** The Recognizing Textual Entailment (RTE) datasets Poliak (2020) come from a series of textual entailment challenges. Data from RTE1, RTE2, RTE3 and RTE5 is combined. The dataset includes 2490 samples for training, 277 for validation, and 3000 for testing. Each data include premise and hypothesis as input prompt and label as generated output. The label is to indicate whether the premise is an entailment for the hypothesis.

For the data above, we first filter out sensitive/toxic content using gpt-3.5-turbo/GPT-4. Second, due to the limited quotas for calling LLMs, for each dataset we randomly select 320 data (277 for RTE, as there are only 277 validation samples in total) from the validation set to conduct our experiments. We do not use the test sets as their annotations are not released, especially BoolQ, QQP, and RTE. As there is no extra training/fine-tuning, the validation set can act as test set in our experiments.

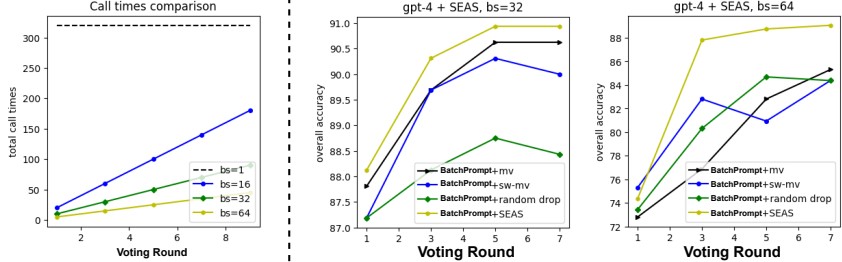

Figure 3: Left: An comparison for the number LLMs calls using BatchPrompt with BPE, with/without SEAS. Right: Ablation study for different modules.

**Language models and prompts:** We evaluate BatchPrompt, as well as BPE over two transformer-based language models with varying scales, i.e., gpt-3.5-turbo and GPT-4. The prompts can be found in 6.

**Parameters:** We perform all experiments in the few-shot setting, without training or fine-tuning the language model. We use 2, 4, and 4 few shot examples for RTE, QQP, BoolQ respectively, which are all selected from training sets with given labels. As had been mentioned in Zhao et al. (2021) that varying the permutation of few-shot examples can cause the accuracy of GPT-3 to range from a low number to near state-of-the-art accuracy, we randomly choose two to four data from the training sets without cherry picking. We also manually assign "confident"/"not confident" to the labels of examples. Temperature is always set to 0 for consistent results. For the comparison, we use similar prompts for different datasets. The batch sizes we use for RTE, QQP, BoolQ are 16/32/64/160 (not for Boolq due to 32k token limit) for GPT-4 whose maximum input token number is 32k, and 16/32 for gpt-3.5-turbo whose maximum input token number is 8k. The number of voting rounds we choose is 1, 3, 5, 7, and 9, which are all odd numbers to avoid situations of tied vote counts.

| bs/vr/model | | gpt-3.5-turbo | | | GPT-4 | | |
|---|---|---|---|---|---|---|---|
| | | mv | sw-mv | sw-mv-neg | mv | sw-mv | sw-mv-neg |
| 16 | 1 | 77.5 | 78.4 | 78.8 | 89.1 | 88.8 | 89.7 |
| | 3 | 80.0 | 81.3 | 78.1 | 89.4 | 89.7 | 89.7 |
| | 5 | 79.7 | 80.3 | 80.3 | 89.1 | 89.7 | 90.3 |
| | 7 | 82.5 | 82.2 | 80.6 | 89.4 | 89.1 | 90.6 |
| | 9 | 81.3 | 81.6 | 80.3 | 89.7 | 89.1 | 90.6 |
| 32 | 1 | 70.0 | 77.2 | 75.3 | 87.8 | 85.6 | 87.8 |
| | 3 | 75.9 | 79.4 | 76.4 | 89.7 | 89.1 | 90.9 |
| | 5 | 77.2 | 80.0 | 78.8 | 90.6 | 89.4 | 89.7 |
| | 7 | 81.0 | 78.2 | 79.1 | 90.6 | 89.4 | 90.6 |
| | 9 | 77.81 | 77.8 | 78.8 | 90.9 | 89.7 | **91.6** |
| 64 | 1 | | | | 72.8 | 76.9 | 75.9 |
| | 3 | | | | 76.8 | 81.3 | 81.9 |
| | 5 | | | | 82.8 | 84.1 | 83.1 |
| | 7 | | | | 85.3 | 86.6 | 85.9 |
| | 9 | | | | 86.3 | 85.9 | 86.6 |
| 1 | 1 | 86.8 | **86.9** | 85.0 | 90.6 | 90.9 | 90.0 |

Table 1: Comparisons of BatchPrompt + BPE on Boolq dataset ("bs" and "vr" are batch size and number of voting rounds). We note that our methods perform better on GPT-4, and suspect that this is because voting is most successful when baseline performance is strong.

## 4.2 RESULTS FOR BATCHPROMPT + BPE

**Complexity comparison:** We use both token count and number of LLM calls to compare the complexity. As can be found in the left part of Fig. 3, LLM calls decrease significantly even with nine voting rounds. The same goes for BatchPrompt with SEAS. Without SEAS, token count increases linearly with increase in voting rounds. For all three tasks, although the token count for BatchPrompt

is only 1/5 of SinglePrompt when the voting round is 1; that token count can even exceed that of SinglePrompt when 9+ voting rounds are used. This also highlights the importance of SEAS from the perspective of token saving. However, we note that BatchPrompt without SEAS cannot always generate better accuracies than BatchPrompt with SEAS. More analysis will be provided below.

**Accuracy comparison:** Results are shown in Tables 1, 7, and 8 (latter two in Appendix). First, prompting with larger batch size generally produces worse results. When batch size is 64, accuracies for all datasets decrease significantly with one voting round, but increase to as high as SinglePrompt with 5+ voting rounds. Second, with more voting rounds, accuracies do not always increase consistently – this is because wrong labels will also accumulate for hard samples. However, when we introduce SEAS, this phenomenon is relieved and accuracies consistently rise with more voting rounds, as can be seen in the next section. Third, BatchPrompt works better on GPT-4 than gpt-3.5-turbo. This is due to the inherent limitation of majority voting. If the accuracy is not good enough as on gpt-3.5-turbo, the wrong prediction will also accumulate with more voting rounds. Therefore, BPE cannot be effective enough. This also gives us a hint that if the general accuracy for a task is quite low (30%), BatchPrompt might not be applicable. Finally, through comparison with BatchPrompt + SEAS, we find BatchPrompt without SEAS can get even better accuracy although with higher token count. For example, the best accuracy is 92.9% for RTE while only 91.7% when SEAS is added. This is because the SEAS decreases the number of voting rounds for easy samples, which might just have been a part of two or three voting rounds. The predicted labels/answers for these easy samples might be wrong.

**Batch size lower than 64:** As can be found in Tables 1-8, accuracy is good for BatchPrompt even with only one voting round, compared with SinglePrompt as shown in the last rows. More voting rounds will increase token count, while the accuracy cannot improve proportionally. Therefore, in this case, if the researcher cares more about efficiency, we encourage researchers to use BatchPrompt without BPE. But when the accuracy is most important, we still encourage more voting rounds, as BatchPrompt + BPE with low batch size might perform even better than SinglePrompt (Boolq: 90.6%-90.9%, RTE: 91.45%-92.9%, QQP: 87.2%-87.8%).

**Batch size equal to or higher than 64:** We can see that the token limits increase dramatically for the latest LLMs, which should be the trend. As shown in the results, we find when batch size is equal to or higher than 64, BatchPrompt cannot perform well without BPE, and accuracy in most cases is much lower than SinglePrompt. For example, the accuracy on Boolq is only 72.8%, while SinglePrompt is 90.6%. However, with more voting rounds, the accuracy of BatchPrompt can smoothly increase to 86.3%, which is competitive with SinglePrompt.

**Negative few-shot examples**: In the tables, we also show results of self-weighted majority voting with negative few-shot samples (sw-mv-neg). We note that if few-shot examples with correct answers all carry "confident" labels, LLMs do not have "not confident" cases and give around 90% "confident" ratings. Therefore, we add two negative few-shot samples to each experiment for completeness. Specifically, the label/answer of the negative few-shot samples will be wrong, followed by a "not confident". We can find in the result tables that "sw-mv-neg" achieves even worse results than "sw-mv". Our conclusion is that although the "not confident" cases are given, the existence of negative few-shot samples will give LLMs a wrong guidance for the labels/answers that will affect LLM judgment more. A better selection of negative few-shot sample might help in our future work.

## 4.3 Results for BatchPrompt + BPE + SEAS

**General Results:** We show the results of BatchPrompt + BPE + SEAS in Fig. 4. We find the number of LLM calls and token count is much lower than SinglePrompt (the black line), as shown in the last two columns. With far fewer tokens, overall accuracies of BatchPrompt + BPE + SEAS are competitive with SinglePrompt. We can also find when SEAS is applied, the token count increases slowly after voting round 3, which means for most easy data, the two consecutive consistent answers requirement has been fulfilled already at very early stage. This is also the reason that SEAS can effectively save tokens.

**Ablation Study:** As in SEAS, from round three, each round will result in a decrease in batch size which might also bring an increase in prediction accuracy. To demonstrate the effectiveness of self-reflection, we compare the results of BatchPrompt + BPE + SEAS (yellow line) with BatchPrompt

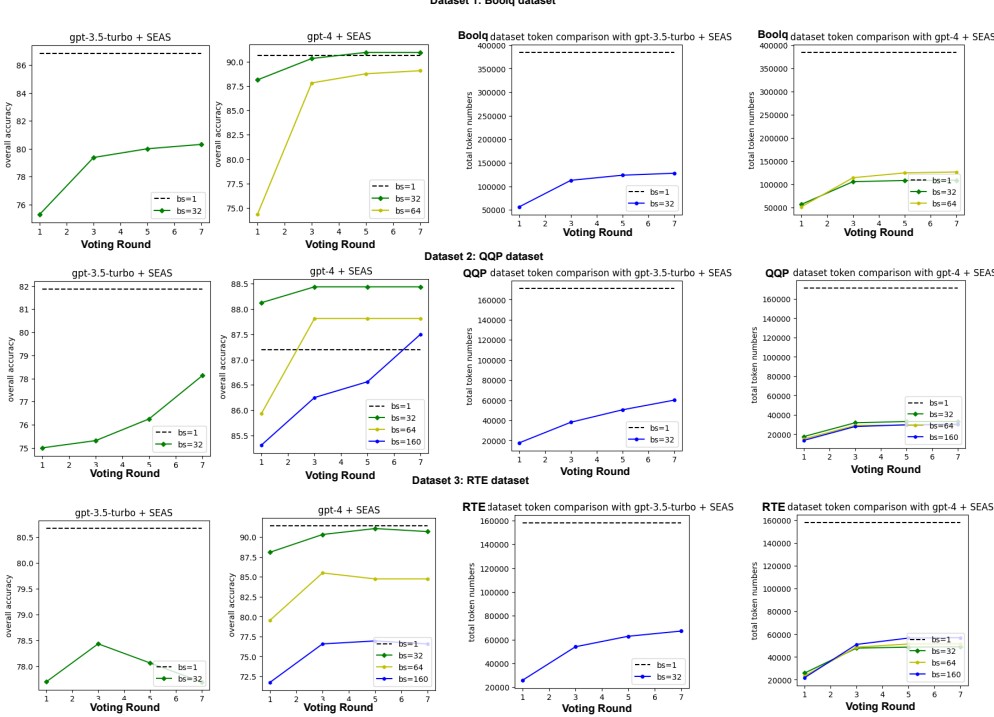

Figure 4: Using SEAS consumes far fewer tokens (two right columns), while producing accuracies near or above baseline SinglePrompt performance (left two columns). This is especially true for GPT-4 (second column) with lower batch sizes. Please notice the black dotted line representing batch size 1 is the result of SinglePrompt without voting. Detailed number can be found in appendix.

+ BPE +random drop (green line). The results comparing accuracies are shown in the right part of Fig. 3. We also include the results of BatchPrompt + BPE as comparison.

## 5 RELATED WORK

A most related work can be found in Cheng et al. (2023), which introduces the idea of naive batching with batch size lower than 6. We noted earlier the significant degradation associated with this technique, we advance several techniques to counteract this to improve performance.

**Large language models prompting:** This paper is mainly related to LLM prompting Petroni et al. (2019). The recent trend of using LLMs is through pre-training, prompting, and prediction Liu et al. (2023b) Radford et al. (2019) Schick & Schütze (2020). This type of method only needs to be given a suite of appropriate prompts, as well as a single language model trained in an entirely unsupervised fashion to solve a great number of tasks Sun et al. (2021b) Brown et al. (2020). The major advantages of these methods are that no extra training/model fine-tuning is needed, and the methods are generally compatible with a wide range of downstream applications. In these methods, prompting is of paramount importance. Considering whether the prompt is human-interpretable, prompt can also be divided into discrete prompt (hard prompt), as well as continuous prompt (soft prompt) Li & Liang (2021). The typical prompt for LLMs is a discrete prompt. There is much research trying to find the best template/format of discrete prompt through paraphrasing Haviv et al. (2021), mining Jiang et al. (2020), Gradient-based Search Wallace et al. (2019) or scoring Davison et al. (2019) etc. A typical discrete prompt for gpt-3.5-turbo/GPT-4 used in the paper is made up of task specification and sample data. Most proposed methods are used to generate better task specifications. Differently, in this paper, our work takes the orthogonal direction, which is batching the sample data to boost prompting performance.

**Long context prompting:** Recent work has demonstrated the length of text will influence the performance of LLMs. In Krishna et al. (2022), it has been demonstrated that long contexts will be more challenging for LLMs. The generation model will fail to condition on the long text which will degrade performance. Another recent work Liu et al. (2023a) also points out that LLMs performance on the multi-document question answering and key-value retrieval tasks will vary according to the text position. The middle part of the input text performs worst. This observation further verifies that when LLMs deal with the long context, the position of the text is significant. One more observation by Sun et al. Sun et al. (2021a) and Sharan et al.Sharan et al. (2018) show that the answer from LLMs for long context come only from the most nearby context together with a set of simple summary statistics for the far away context.

**Consistency in LLMs:** Researchers have gradually found that one of the best methods to improve the performance of LLMs is through repeatedly calling LLMs to generate consistent answers. In Imani et al. (2023), the authors propose MathPrompter which uses multiple ChatGPT calls to generate solutions for math problems Cobbe et al. (2021b), and select the most voted one. Similarly, in Wang et al. (2022), self-consistency is proposed to generate consistent answers for different tasks to boost the performance of LLMs, which is a follow-up work for a chain of thought (COT) Wei et al. (2022). Another follow-up work for COT using self-consistency is Tree-of-Thought Yao et al. (2023), in which multiple reasoning paths are generated while the consistency is achieved through DFS/BFS, instead of simple voting strategy Davani et al. (2022)Chen et al. (2022)Zhou et al. (2002). The basic idea of both methods is that the same problem can be solved in different ways, and the right answer should have the largest possibility to achieve consistency. This idea is mainly borrowed from Nye et al. (2021), in which the consistency of only two systems is used to generate better performance for text generation, and the system is named dual-system. Self-consistency has also been used in other applications, including code generation Chen et al. (2023b), chatbot design Adiwardana et al. (2020), explanation generation Camburu et al. (2019), and knowledge extraction Elazar et al. (2021). An inherent limitation of these consistency-based methods is the increase of time complexity, input token count, number of LLM calls, and cost. It can never be an efficient way to repeat the same problem many times to generate the right answer. In our paper, we for the first time discuss the efficiency problem, and use batching to improve language model efficiency.

## 6 FUTURE WORK

We believe batch prompting should be the trend, with the ever-increasing token limits of LLMs. Although the naive batch prompting way had been noticed before in Cheng et al. (2023), to the best of our knowledge, our work represents the first formal analysis of prompt engineering focused not on expanding context for higher precision on a *single* task, but on expanding the model's workload in the most efficient and performant manner possible. We believe this line of research will be increasingly important, considering the cost and power consumption that comes with increased use.

Despite finding stable strategies to increase efficiency and performance through voting, voting with confidence, permutation, and varied batch sizes; we understand that such strategies must be tuned for each use case, depending on the length of task description, variable performance across individual tasks, and specific LLM used. We leave for future work the task to automate such designs. For example, to implement learning environments that discover the optimal batch size, number of votes, and confidence weighting, for an arbitrary model, context size, text domain, and task set. A reinforcement learning or Bayesian optimization approach might sample from the joint space of batch size, number of votes, and confidence weighting, to find optimal policies and combinations of settings, to reduce calls, tokens, and overall cost. Also, given the boost in performance due to confidence weights, we suspect that full exposition of this strategy is possible in future work.

We believe the recently proposed approach of optimally selecting among a set of LLMs to be another useful strategy for efficiency Chen et al. (2023a), and could be combined with our BatchPrompt.

Finally, BatchPrompt is mainly used for three tasks which were transformed to binary classification tasks in huggingface, as well as arithmetic reasoning tasks. However, we believe the BatchPrompt + BPE + SEAS framework is also applicable to other NLP tasks, if the task is to generate a specific answer (e.g., machine translation, essay grading, etc.). The framework might not be applicable for text summarization or generation, for which answers vary largely in repeated experiments (voting rounds). Application of the proposed framework on these additional NLP tasks is left as future work.

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

## A    DETAILS FOR FIG.4

The detailed numbers for Fig. 4 can be found in Table 2, 3, and 4.

| bs/vt/model | | gpt-3.5-turbo+BPE+SEAS | | gpt-4+BPE+SEAS | |
|---|---|---|---|---|---|
| | | Accuracy | Token | Accuracy | Token |
| 32 | 1 | 75.31 | 56346 | 88.13 | 56346 |
| | 3 | 79.38 | 112780 | 90.31 | 105419 |
| | 5 | 80.00 | 123623 | 90.94 | 107957 |
| | 7 | 80.30 | 127807 | 90.94 | 108490 |
| 64 | 1 | | | 74.38 | 51046 |
| | 3 | | | 87.81 | 113933 |
| | 5 | | | 88.75 | 124424 |
| | 7 | | | 89.06 | 126337 |
| 1 | 1 | 86.83 | 384946 | 90.63 | 384946 |

Table 2: Comparisons of BatchPrompt + BPE + SEAS; on Boolq dataset ("bs" and "vr" are batch size and number of voting rounds).

| bs/vt/model | | gpt-3.5-turbo+BPE+SEAS | | gpt-4+BPE+SEAS | |
|---|---|---|---|---|---|
| | | Accuracy | Token | Accuracy | Token |
| 32 | 1 | 75.00 | 17641 | 88.13 | 17641 |
| | 3 | 75.31 | 38148 | 88.44 | 31900 |
| | 5 | 76.25 | 50514 | 88.44 | 33170 |
| | 7 | 78.13 | 60133 | 88.44 | 33253 |
| 64 | 1 | | | 85.94 | 15161 |
| | 3 | | | 87.81 | 29058 |
| | 5 | | | 87.81 | 29509 |
| | 7 | | | 87.81 | 29581 |
| 160 | 1 | | | 85.31 | 13673 |
| | 3 | | | 86.25 | 27987 |
| | 5 | | | 86.56 | 29765 |
| | 7 | | | 87.50 | 30396 |
| 1 | 1 | 81.88 | 171401 | 87.2 | 171401 |

Table 3: Comparisons of BatchPrompt + BPE + SEAS; on QQP dataset ("bs" and "vr" are batch size and number of voting rounds).

## B    EXTENSION TO ARITHMETIC REASONING TASK

**GSM8K:** The Graduate School Math 8K is a dataset of 8.5K high-quality linguistically diverse grade school math word problems created by human problem writers Cobbe et al. (2021a). The dataset includes 7.5K training problems and 1K test problems. It will take between 2 and 8 steps to solve these problems. Each data include a question and an answer. We use the whole testing set with 1280 filtered samples for GSM8K for inference, as there is no validation set for this data, and the testing set is fully annotated. We use two few-shot examples for this dataset. We do not do a large number of comparisons on GSM8K due to the current limit of token numbers. As the chain-of-thought reasoning needs long text, we can only use GPT-4 (32k tokens) for this task and have to keep the batch size lower than 32.

**Results:** We also demonstrate the proposed BatchPrompt on the GSM8K dataset. The accuracy of SinglePrompt is 94.9% (ranks 5th on the leaderboard). When batch size is 32, accuracy changes from 89.1% (1 voting round) to 92.0% (5 voting rounds) with the increase of voting rounds. We do not use SEAS for this side experiment, our observation is that accuracy of large batch size is already very high even using only one voting round. This observation might be caused by the existence of new "few-shot samples" when the batch size is larger than 1. In a batch, the former data sample can

| bs/vt/model | | gpt-3.5-turbo+BPE+SEAS | | gpt-4+BPE+SEAS | |
|---|---|---|---|---|---|
| | | Accuracy | Token | Accuracy | Token |
| 32 | 1 | 77.70 | 25921 | 88.10 | 25921 |
| | 3 | 78.43 | 54000 | 90.33 | 47807 |
| | 5 | 78.06 | 62838 | 91.08 | 48693 |
| | 7 | 77.69 | 67191 | 90.70 | 48970 |
| 64 | 1 | | | 79.55 | 23366 |
| | 3 | | | 85.50 | 48647 |
| | 5 | | | 84.75 | 51420 |
| | 7 | | | 84.75 | 51541 |
| 160 | 1 | | | 71.75 | 21833 |
| | 3 | | | 76.58 | 51048 |
| | 5 | | | 76.95 | 56633 |
| | 7 | | | 76.58 | 57004 |
| 1 | 1 | 80.67 | 158270 | 91.45 | 158270 |

Table 4: Comparisons of BatchPrompt + BPE + SEAS; on RTE dataset ("bs" and "vr" are batch size and number of voting rounds).

actually be viewed as few-shot samples for the later ones. When the accuracy is high (90%+), LLMs can generate correct answers for these "few-shot samples", which could give good guidance for the later samples. As this arithmetic reasoning task relies heavily on few-shot logical deduction, more "few-shot samples" can lead to a better result.

## C  EXTENSION TO CAUSAL REASONING AND NATURAL LANGUAGE INFERENCE TASKS

To increase confidence, we include the following raw results on additional datasets, that have long answers, and where all validation data (>300) is used.

**COPA (Choice of Plausible Alternatives) Roemmele et al. (2011)**: See Table 5.

**Multi-Genre Natural Language Inference (MNLI) Williams et al. (2017)**: See Table 6

## D  PROMPTS

We want to mention that the batched input must be in format of "Data 1, Data 2...". Not using the index or just use a delimiter "—" will result in the missing of answers (e.g., 15 generated answers when input batch size is 16). This is also the reason that we give an extra "batch size" reminder at the end of each prompt.

[Conf-Description]: 'You not only need to generate The label/answer, but also your confidence. If you are confident in your output class, append a "(confident)" at the end of the label; else, append a "(not confident)".'

[Place-Holder-Conf]: '(confident or not confident)'

If we are using Self weighted Majority Voting, the above two place holder will be added; else, we will be using regular Majority Voting and the two place holder will be NONE.

**Prompts for Boolq:**

You are a professional NLP expert at Question Answering annotation. Please generate labels given instructions. You will be given [BATCH-SIZE] passages with questions each time, as input.

Each input includes a 'passage' and a 'question' about the passage.

Your goal to determine whether the answer to the question is yes or no and classify, as below:

[class 0]: if the answer is 'No'.

| gpt-3.5-turbo Accuracy | VT=1 | VT=3 | VT=5 | VT=7 | VT=9 |
|---|---|---|---|---|---|
| BS=1 | 89.0625 | - | - | - | - |
| BS=16 | 82.8125 | 82.8125 | 85.5 | 85.9375 | 85.9375 |
| BS=32 | 70.3125 | 76.5625 | 79.6875 | 85.9375 | 85.9375 |
| gpt-3.5-turbo Token Num | VT=1 | VT=3 | VT=5 | VT=7 | VT=9 |
| BS=1 | 37466 | - | - | - | - |
| BS=16 | 4646 | 9642 | 9833 | 10112 | 10112 |
| BS=32 | 3552 | 7961 | 8324 | 8525 | 8525 |
| gpt-3.5-turbo Calling Num | VT=1 | VT=3 | VT=5 | VT=7 | VT=9 |
| BS=1 | 64 | - | - | - | - |
| BS=16 | 4 | 12 | 20 | 28 | 36 |
| BS=32 | 2 | 6 | 10 | 14 | 18 |
| GPT-4 Accuracy | VT=1 | VT=3 | VT=5 | VT=7 | VT=9 |
| BS=1 | 96.875 | - | - | - | - |
| BS=16 | 98.4375 | 98.4375 | 98.4375 | 98.4375 | 98.4375 |
| BS=32 | 98.4375 | 98.4375 | 98.4375 | 98.4375 | 98.4375 |
| GPT-4 Token Num | VT=1 | VT=3 | VT=5 | VT=7 | VT=9 |
| BS=1 | 37466 | - | - | - | - |
| BS=16 | 4646 | 9862 | 9862 | 9862 | 9862 |
| BS=32 | 3552 | 8468 | 8468 | 8468 | 8468 |
| GPT-4 Calling Num | VT=1 | VT=3 | VT=5 | VT=7 | VT=9 |
| BS=1 | 64 | - | - | - | - |
| BS=16 | 4 | 12 | 20 | 28 | 36 |
| BS=32 | 2 | 6 | 10 | 14 | 18 |

Table 5: gpt-3.5-turbo and GPT4 Results on COPA

| gpt-3.5-turbo Accuracy | VT=1 | VT=3 | VT=5 | VT=7 | VT=9 |
|---|---|---|---|---|---|
| BS=1 | 77.5% | - | - | - | - |
| BS=4 | 66.3% | 68.2% | 72.1% | 72.7% | 72.7% |
| BS=16 | 64.2% | 63.4% | 71.2% | 71.6% | 72.3% |
| gpt-3.5-turbo Token Num | VT=1 | VT=3 | VT=5 | VT=7 | VT=9 |
| BS=1 | 158401 | - | - | - | - |
| BS=4 | 51361 | 82723 | 90085 | 90447 | 90447 |
| BS=16 | 24601 | 55963 | 62325 | 64435 | 65524 |
| gpt-3.5-turbo Calling Num | VT=1 | VT=3 | VT=5 | VT=7 | VT=9 |
| BS=1 | 320 | - | - | - | - |
| BS=4 | 80 | 240 | 400 | 560 | 720 |
| BS=16 | 20 | 60 | 100 | 140 | 180 |
| GPT-4 Accuracy | VT=1 | VT=3 | VT=5 | VT=7 | VT=9 |
| BS=1 | 88.8% | - | - | - | - |
| BS=64 | 75.3% | 80.3% | 82.8% | 82.2% | 83.1% |
| GPT-4 Token Num | VT=1 | VT=3 | VT=5 | VT=7 | VT=9 |
| BS=1 | 152321 | - | - | - | - |
| BS=64 | 17816 | 49178 | 54540 | 54582 | 54668 |
| GPT-4 Calling Num | VT=1 | VT=3 | VT=5 | VT=7 | VT=9 |
| BS=1 | 320 | - | - | - | - |
| BS=64 | 5 | 15 | 25 | 35 | 45 |

Table 6: gpt-3.5-turbo and GPT-4 Results on MNLI

[class 1]: if the answer is 'Yes'.

Given a input, please output a label ([class 0] or [class 1]) [Conf-Description].

You will be given [BATCH-SIZE] inputs each time, and the below is the format of input which will be given:

============

Input 0: xxxxx

Input 1: xxxxx

......

============

Below are the outputs you need to generate. "X" can be '0' or '1'.

============

Label for Input 0: [class X] [Place-Holder-Conf]

Label for Input 1: [class X] [Place-Holder-Conf]

......

============

Please make sure each generated label is in format of [class X].

Please make sure to generate [BATCH-SIZE] labels. You may include other additional sections here.

**Prompts for QQP:**

You are a professional NLP expert at duplicate question detection. You will be given [BATCH-SIZE] pairs of data from Quora Question Pairs (QQP) dataset each time, as input. Each data includes a pair data, "Question1" and "Question2". Your goal to determine whether two questions are duplicates of each other. You need to classify into below two classes:

[class 1]: if they have the same meaning (semantically equivalent).

[class 0]: if they do NOT have the same meaning.

You will be given [BATCH-SIZE] question pairs each time, and the below is the format of question pairs which will be given:

============

Question pair 0:

Question1: xxxxx

Question2: xxxxx

Question pair 1:

Question1: xxxxx

Question2: xxxxx

......

============

Below are the outputs you need to generate. "X" can be '1' or '0'. [Conf-Description]

============

Label for Question pair 0: [class X][Place-Holder-Conf]

Label for Question pair 1: [class X][Place-Holder-Conf]

......

============

Please make sure each generated label is in format of [class X].

Please make sure to generate [BATCH-SIZE] labels.

**Prompts for RTE:**

You are a professional NLP expert at sentence pair relationship annotation. You will be given [BATCH-SIZE] sentence pairs from Textual Entailment Recognition dataset each time, as input. Each data includes a sentence pair, "Premise" and "Hypothesis". Your goal is to classify the sentence pair into two classes as below:

[class 0]: the given Hypothesis and Premise are logical and following (entailment) to each other.

[class 1]: the given Hypothesis and Premise are NOT following (entailment) to each other.

You will be given [BATCH-SIZE] sentence pairs each time, and the below is the format of sentence pairs which will be given:

============

Sentence pair 0:

Premise: xxxxx

Hypothesis: xxxxx

Sentence pair 1:

Premise: xxxxx

Hypothesis: xxxxx

......

============

Below are the outputs you need to generate. "X" can be '1' or '0'. [Conf-Description]

============

Label for Sentence pair 0: [class X][Place-Holder-Conf]

Label for Sentence pair 1: [class X][Place-Holder-Conf]

......

============

Please make sure each generated label is in format of [class X].

Please make sure to generate [BATCH-SIZE] labels.

**Prompts for GSM8K:**

You will be given [BATCH-SIZE] math probelms.

These problems take between 2 and 8 steps to solve, and solutions primarily involve performing a sequence of elementary calculations using basic arithmetic operations to reach the final answer.

The below is the format of sentence pairs which will be given:

============

Input 0: xxxxx

Input 1: xxxxx

......

============

Below are the calculation results you need to generate. [Conf-Description]

============ Result for Input 0: [intermediate reasoning steps], The answer is xxxx. [Placeholder-Conf]

Result for Input 1: [intermediate reasoning steps], The answer is xxxx. [Place-holder-Conf]

......

============

Please make sure to write your a series of intermediate reasoning steps. Please make sure the final sentence is "The answer is xxx.", and the answer should be a number.

Please make sure to generate [BATCH-SIZE] labels each time.

## E  DATA INDEX

To save LLMs calls, we use the gpt-3.5-turbo to filter out the data containing sensitive content that cannot be used in LLMs, and randomly choose 320 samples for Boolq, QQP, RTE, 1280 samples for GSM8K. There's no bias for data selections.

**Validation set on Boolq:** 1538, 220, 1371, 2128, 1481, 1822, 37, 2696, 1276, 2292, 1057, 2537, 1490, 542, 3069, 1387, 1, 1780, 2788, 1930, 2807, 2918, 254, 2636, 2515, 3007, 1285, 909, 1973, 2894, 3178, 2629, 284, 460, 968, 774, 2280, 1650, 703, 1951, 2544, 439, 3263, 2993, 491, 804, 257, 1340, 1948, 2308, 994, 1579, 2350, 1574, 2834, 338, 2773, 1909, 2136, 1396, 2685, 2821, 2996, 1422, 2110, 850, 584, 1063, 1464, 3157, 1749, 671, 358, 2820, 659, 1509, 2574, 893, 3135, 798, 296, 2655, 2726, 1471, 3016, 2545, 2316, 336, 1769, 2732, 692, 1073, 2922, 2328, 322, 1784, 10, 2852, 3249, 2513, 951, 2631, 2677, 569, 127, 1913, 3184, 2389, 466, 167, 1857, 2327, 1448, 1322, 2242, 1275, 860, 2038, 918, 2707, 237, 1994, 1933, 2484, 442, 1287, 124, 1150, 429, 1684, 462, 416, 2660, 1494, 2891, 651, 2237, 1302, 51, 438, 1955, 986, 626, 140, 269, 2247, 2057, 2461, 1190, 1953, 3063, 783, 444, 2017, 886, 817, 79, 1567, 1314, 3118, 2329, 2806, 2602, 1426, 3174, 2647, 2623, 3096, 2674, 1090, 819, 1144, 1334, 3254, 669, 1794, 2609, 149, 1990, 313, 376, 690, 2854, 1935, 1732, 796, 2000, 2659, 1293, 294, 3086, 311, 827, 2510, 364, 1115, 2494, 1868, 305, 1670, 2433, 2348, 1700, 878, 1816, 2340, 1678, 56, 1069, 2747, 2905, 3182, 506, 691, 110, 2075, 1778, 2001, 1677, 3119, 544, 2653, 3143, 3004, 14, 2274, 1354, 1175, 1243, 70, 137, 2955, 177, 578, 2204, 496, 88, 1737, 563, 611, 2808, 621, 2421, 1456, 2163, 3122, 2770, 1828, 2753, 2373, 1172, 1030, 3169, 663, 1153, 956, 2332, 1171, 1429, 805, 1380, 3136, 1375, 2751, 1811, 2758, 277, 2931, 557, 813, 1008, 1028, 966, 1048, 2995, 1562, 2152, 1587, 108, 3209, 2692, 2671, 172, 652, 387, 1155, 3066, 1906, 2591, 1814, 349, 1774, 2141, 1232, 2422, 335, 2801, 1922, 1968, 1755, 2317, 881, 2524, 2884, 2558, 2290, 2272, 2330, 1734, 2621, 1239, 115, 3176, 2407, 2425, 1596, 1854, 1839, 818

**Validation set on QQP:** 0, 1, 2, 3, 4, 5, 6, 7, 8, 9, 10, 11, 12, 13, 14, 15, 16, 17, 18, 19, 20, 21, 22, 23, 24, 25, 26, 27, 28, 29, 30, 31, 32, 33, 34, 35, 36, 37, 38, 39, 40, 41, 42, 43, 44, 45, 46, 47, 48, 49, 50, 51, 52, 53, 54, 55, 56, 57, 58, 59, 60, 61, 62, 63, 64, 65, 66, 67, 68, 69, 70, 71, 72, 73, 74, 75, 76, 77, 78, 79, 80, 81, 82, 83, 84, 85, 86, 87, 88, 89, 90, 91, 92, 93, 94, 95, 96, 97, 98, 99, 100, 101, 102, 103, 105, 106, 107, 108, 109, 110, 111, 112, 113, 114, 115, 116, 117, 118, 119, 120, 121, 122, 123, 124, 125, 126, 127, 128, 129, 130, 132, 133, 134, 135, 136, 137, 138, 139, 140, 141, 142, 143, 144, 145, 146, 147, 148, 149, 150, 151, 152, 153, 154, 155, 156, 157, 158, 159, 160, 161, 162, 163, 164, 165, 166, 167, 168, 169, 170, 171, 172, 173, 174, 175, 176, 177, 178, 179, 180, 181, 182, 183, 184, 185, 186, 187, 188, 189, 190, 191, 192, 193, 194, 195, 196, 197, 198, 199, 200, 201, 202, 204, 205, 206, 207, 208, 209, 210, 211, 212, 213, 214, 215, 216, 217, 218, 219, 220, 221, 222, 223, 224, 225, 226, 227, 228, 229, 230, 232, 233, 234, 235, 236, 237, 238, 239, 240, 241, 242, 243, 244, 245, 246, 247, 248, 249, 250, 251, 252, 253, 254, 255, 256, 257, 258, 259, 260, 261, 262, 263, 264, 265, 266, 267, 268, 269, 270, 271, 272, 273, 275, 276, 277, 278, 279, 280, 281, 282, 283, 284, 285, 286, 287, 288, 289, 290, 291, 292, 293, 294, 295, 296, 297, 298, 299, 300, 301, 302, 303, 304, 305, 306, 307, 308, 309, 310, 311, 312, 313, 314, 315, 316, 317, 318, 319, 320, 321, 322, 323, 324

**Validation set on RTE:** 0, 1, 2, 3, 4, 5, 6, 7, 8, 9, 10, 11, 12, 13, 14, 15, 16, 17, 18, 19, 20, 22, 23, 24, 25, 26, 27, 28, 29, 30, 32, 33, 34, 35, 36, 37, 38, 39, 40, 41, 42, 43, 44, 45, 46, 47, 48, 49, 50, 51, 53, 54, 55, 57, 58, 59, 60, 61, 62, 63, 64, 65, 66, 67, 68, 69, 70, 71, 72, 73, 74, 75, 76, 77, 78, 79, 80, 81, 82, 83, 84, 85, 86, 87, 88, 89, 90, 91, 92, 93, 94, 95, 96, 97, 98, 99, 100, 101, 102, 103, 104, 105, 106, 107, 108, 109, 110, 111, 112, 113, 114, 115, 116, 117, 118, 119, 120, 121, 122, 123, 124, 125, 126, 127, 128, 129, 130, 131, 132, 133, 134, 135, 136, 137, 138, 139, 140, 141, 143, 144, 145, 146, 147, 148, 149, 150, 151, 152, 153, 154, 155, 156, 157, 158, 159, 160, 161, 162, 163, 165, 166, 167, 168, 169, 170, 171, 172, 173, 174, 175, 176, 177, 178, 179, 180, 181, 182, 183, 184, 185, 186, 187, 188, 189, 190, 191, 192, 193, 195, 196, 197, 198, 199, 200, 201, 202, 203, 204, 205, 206, 207, 208, 209, 210, 211, 212, 213, 214, 215, 216, 218, 219, 220, 221, 222, 223, 224, 225, 226, 227,

228, 229, 230, 231, 232, 233, 234, 235, 236, 237, 238, 239, 240, 241, 242, 243, 244, 245, 246, 247, 248, 249, 250, 251, 252, 253, 254, 255, 256, 257, 258, 259, 260, 261, 262, 263, 264, 265, 266, 267, 268, 269, 270, 271, 272, 273, 274, 275, 276

**Testing set on GSM8K(1319 in total, 39 filtered out):** 0, 1, 2, 3, 4, 5, 6, 7, 8, 9, 10, 11, 12, 13, 14, 15, 16, 17, 18, 19, 20, 21, 22, 23, 24, 25, 26, 27, 28, 29, 30, 31, 32, 33, 34, 35, 36, 37, 38, 39, 40, 41, 42, 43, 44, 45, 46, 47, 48, 49, 50, 51, 52, 53, 54, 55, 56, 57, 58, 59, 60, 61, 62, 63, 64, 65, 66, 67, 68, 69, 70, 71, 72, 73, 74, 75, 76, 77, 78, 79, 80, 81, 82, 83, 84, 85, 86, 87, 88, 89, 90, 91, 92, 93, 94, 95, 96, 97, 98, 99, 100, 101, 102, 103, 104, 105, 106, 107, 108, 109, 110, 111, 112, 113, 114, 115, 116, 117, 118, 119, 120, 121, 122, 123, 124, 125, 126, 127, 128, 129, 130, 131, 132, 133, 134, 135, 136, 137, 138, 139, 140, 141, 142, 143, 144, 145, 146, 147, 148, 149, 150, 151, 152, 153, 154, 155, 156, 157, 158, 159, 160, 161, 162, 163, 164, 165, 166, 167, 168, 169, 170, 171, 172, 173, 174, 175, 176, 177, 178, 179, 180, 181, 182, 183, 184, 185, 186, 187, 188, 189, 190, 191, 192, 193, 194, 195, 196, 197, 198, 199, 200, 201, 202, 203, 204, 205, 206, 207, 208, 209, 210, 211, 212, 213, 214, 215, 216, 217, 218, 219, 220, 221, 222, 223, 224, 225, 226, 227, 228, 229, 230, 231, 232, 233, 234, 235, 236, 237, 238, 239, 240, 241, 242, 243, 244, 245, 246, 247, 248, 249, 250, 251, 252, 253, 254, 255, 256, 257, 258, 259, 260, 261, 262, 263, 264, 265, 266, 267, 268, 269, 270, 271, 272, 273, 274, 275, 276, 277, 278, 279, 280, 281, 282, 283, 284, 285, 286, 287, 288, 289, 290, 291, 292, 293, 294, 295, 296, 297, 298, 299, 300, 301, 302, 303, 304, 305, 306, 307, 308, 309, 310, 311, 312, 313, 314, 315, 316, 317, 318, 319, 320, 321, 322, 323, 324, 325, 326, 327, 328, 329, 330, 331, 332, 333, 334, 335, 336, 337, 338, 339, 340, 341, 342, 343, 344, 345, 346, 347, 348, 349, 350, 351, 352, 353, 354, 355, 356, 357, 358, 359, 360, 361, 362, 363, 364, 365, 366, 367, 368, 369, 370, 371, 372, 373, 374, 375, 376, 377, 378, 379, 380, 381, 382, 383, 384, 385, 386, 387, 388, 389, 390, 391, 392, 393, 394, 395, 396, 397, 398, 399, 400, 401, 402, 403, 404, 405, 406, 407, 408, 409, 410, 411, 412, 413, 414, 415, 416, 417, 418, 419, 420, 421, 422, 423, 424, 425, 426, 427, 428, 429, 430, 431, 432, 433, 434, 435, 436, 437, 438, 439, 440, 441, 442, 443, 444, 445, 446, 447, 448, 449, 450, 451, 452, 453, 454, 455, 456, 457, 458, 459, 460, 461, 462, 463, 464, 465, 466, 467, 468, 469, 470, 471, 472, 473, 474, 475, 476, 477, 478, 479, 480, 481, 482, 483, 484, 485, 486, 487, 488, 489, 490, 491, 492, 493, 494, 495, 496, 497, 498, 499, 500, 501, 502, 503, 504, 505, 506, 507, 508, 509, 510, 511, 512, 513, 514, 515, 516, 517, 518, 519, 520, 521, 522, 523, 524, 525, 526, 527, 528, 529, 530, 531, 532, 533, 534, 535, 536, 537, 538, 539, 540, 541, 542, 543, 544, 545, 546, 547, 548, 549, 550, 551, 552, 553, 554, 555, 556, 557, 558, 559, 560, 561, 562, 563, 564, 565, 566, 567, 568, 569, 570, 571, 572, 573, 574, 575, 576, 577, 578, 579, 580, 581, 582, 583, 584, 585, 586, 587, 588, 589, 590, 591, 592, 593, 594, 595, 596, 597, 598, 599, 600, 601, 602, 603, 604, 605, 606, 607, 608, 609, 610, 611, 612, 613, 614, 615, 616, 617, 618, 619, 620, 621, 622, 623, 624, 625, 626, 627, 628, 629, 630, 631, 632, 633, 634, 635, 636, 637, 638, 639, 640, 641, 642, 643, 644, 645, 646, 647, 648, 649, 650, 651, 652, 653, 654, 655, 656, 657, 658, 659, 660, 661, 662, 663, 664, 665, 666, 667, 668, 669, 670, 671, 672, 673, 674, 675, 676, 677, 678, 679, 680, 681, 682, 683, 684, 685, 686, 687, 688, 689, 690, 691, 692, 693, 694, 695, 696, 697, 698, 699, 700, 701, 702, 703, 704, 705, 706, 707, 708, 709, 710, 711, 712, 713, 714, 715, 716, 717, 718, 719, 720, 721, 722, 723, 724, 725, 726, 728, 729, 730, 731, 732, 733, 734, 735, 736, 737, 738, 739, 740, 741, 742, 743, 744, 745, 746, 747, 748, 749, 750, 751, 752, 753, 754, 755, 756, 757, 758, 759, 760, 761, 762, 763, 764, 765, 766, 767, 768, 769, 770, 771, 772, 773, 774, 775, 776, 777, 778, 779, 780, 781, 782, 783, 784, 785, 786, 787, 788, 789, 790, 791, 792, 793, 794, 795, 796, 797, 798, 799, 800, 801, 802, 803, 804, 805, 806, 807, 808, 809, 810, 811, 812, 813, 814, 815, 816, 817, 818, 819, 820, 821, 822, 823, 824, 825, 826, 827, 828, 829, 830, 831, 832, 833, 834, 835, 836, 837, 838, 839, 840, 841, 842, 843, 844, 845, 846, 847, 848, 849, 850, 851, 852, 853, 854, 855, 856, 857, 858, 859, 860, 861, 862, 863, 864, 865, 866, 867, 868, 869, 870, 871, 872, 873, 874, 875, 876, 877, 878, 879, 880, 881, 882, 883, 884, 885, 886, 887, 888, 889, 890, 891, 892, 893, 894, 895, 896, 897, 898, 899, 900, 901, 902, 903, 904, 905, 906, 907, 908, 909, 910, 911, 912, 913, 914, 915, 916, 917, 918, 919, 920, 921, 922, 923, 924, 925, 926, 927, 928, 929, 930, 931, 932, 933, 934, 935, 936, 937, 938, 939, 940, 941, 942, 943, 944, 945, 946, 947, 948, 949, 950, 951, 952, 953, 954, 955, 956, 957, 958, 959, 960, 961, 962, 963, 964, 965, 966, 967, 968, 969, 970, 971, 972, 973, 974, 975, 976, 977, 978, 979, 980, 981, 982, 983, 984, 985, 986, 987, 988, 989, 990, 991, 992, 993, 994, 995, 996, 997, 998, 999, 1000, 1001, 1002, 1003, 1004, 1005, 1006, 1007, 1008, 1009, 1010, 1011, 1012, 1013, 1014, 1015, 1016, 1017, 1018, 1019, 1020, 1021, 1022, 1023, 1024, 1025, 1026, 1027, 1028, 1029, 1030, 1031, 1032, 1033, 1034, 1035, 1036, 1037, 1038, 1039, 1040, 1041, 1042, 1043, 1044, 1045, 1046, 1047, 1048, 1049, 1050, 1051, 1052, 1053, 1054, 1055, 1056, 1057, 1058, 1059, 1060, 1061, 1062, 1063, 1064, 1065, 1066, 1067, 1068, 1069, 1070, 1071, 1072, 1073, 1074, 1075, 1076, 1077, 1078, 1079, 1080, 1081, 1082, 1083, 1084, 1085, 1086, 1087, 1088, 1089, 1090, 1091,

1092, 1093, 1094, 1095, 1096, 1097, 1098, 1099, 1100, 1101, 1102, 1103, 1104, 1105, 1106, 1107, 1108, 1109, 1110, 1111, 1112, 1113, 1114, 1115, 1116, 1117, 1118, 1119, 1120, 1121, 1122, 1123, 1124, 1125, 1126, 1127, 1128, 1129, 1130, 1131, 1132, 1133, 1134, 1135, 1136, 1137, 1138, 1139, 1140, 1141, 1142, 1143, 1144, 1145, 1146, 1147, 1148, 1149, 1150, 1151, 1152, 1153, 1154, 1155, 1156, 1157, 1158, 1159, 1160, 1161, 1162, 1163, 1164, 1165, 1166, 1167, 1168, 1169, 1170, 1171, 1172, 1173, 1174, 1175, 1176, 1177, 1178, 1179, 1180, 1181, 1182, 1183, 1184, 1185, 1186, 1187, 1188, 1189, 1190, 1191, 1192, 1193, 1194, 1195, 1196, 1197, 1198, 1199, 1200, 1201, 1202, 1203, 1204, 1205, 1206, 1207, 1208, 1209, 1210, 1211, 1212, 1213, 1214, 1215, 1216, 1217, 1218, 1219, 1220, 1221, 1222, 1223, 1224, 1225, 1226, 1227, 1228, 1229, 1230, 1231, 1232, 1233, 1234, 1235, 1236, 1237, 1238, 1239, 1240, 1241, 1242, 1243, 1244, 1245, 1246, 1247, 1248, 1249, 1250, 1251, 1252, 1253, 1254, 1255, 1256, 1257, 1258, 1259, 1260, 1261, 1262, 1263, 1264, 1265, 1266, 1267, 1268, 1269, 1270, 1271, 1272, 1273, 1274, 1275, 1276, 1277, 1278, 1279, 1280

## F EXPERIMENTAL RESULT OF BATCH GPT+BPE ON RTE AND QQP

Results can be found in Table. 7 and Table. 8.

| bs/vr/model | | gpt-3.5-turbo | | | GPT-4 | | |
|---|---|---|---|---|---|---|---|
| | | mv | sw-mv | sw-mv-neg | mv | sw-mv | sw-mv-neg |
| 16 | 1 | 71.8 | 77.3 | 73.6 | 90.3 | 90.0 | 90.7 |
| | 3 | 77.0 | 81.4 | 75.8 | **92.9** | 91.1 | 91.5 |
| | 5 | 77.7 | 79.2 | 78.8 | 92.6 | 91.1 | 90.7 |
| | 7 | 75.8 | 80.3 | 79.2 | 91.8 | 91.1 | 91.8 |
| | 9 | 78.8 | 80.3 | 79.9 | 91.5 | 90.7 | 91.5 |
| 32 | 1 | 71.8 | 75.5 | 74.0 | 88.9 | 88.1 | 89.2 |
| | 3 | 70.3 | 78.4 | 73.6 | 88.1 | 88.5 | 90.0 |
| | 5 | 72.5 | 79.2 | 75.1 | 88.9 | 90.7 | 90.3 |
| | 7 | 72.1 | 79.2 | 74.0 | 90.3 | 90.7 | 90.3 |
| | 9 | 71.8 | 79.2 | 75.5 | 90.0 | 90.3 | 90.7 |
| 64 | 1 | | | | 80.7 | 82.5 | 78.8 |
| | 3 | | | | 85.5 | 85.9 | 86.6 |
| | 5 | | | | 87.7 | 88.5 | 90.7 |
| | 7 | | | | 88.5 | 88.8 | 90.7 |
| | 9 | | | | 88.1 | 89.2 | 90.3 |
| 1 | 1 | **84.4** | 84.75 | 84.0 | 91.4 | 91.1 | 91.1 |

Table 7: Comparisons on RTE dataset.

## G EXAMPLES FOR NEGATIVE FEW-SHOT SAMPLES (GSM8K, BOOLQ)

**GSM8K:**

Input 0: Question: Roger has 5 tennis balls. He buys 2 more cans of tennis balls. Each can has 3 tennis balls. How many tennis balls does he have now?

Input 1: Question: Roger has 5 tennis balls. He buys 2 more cans of tennis balls. Each can has 3 tennis balls. How many tennis balls does he have now?

Input 2: Question: The cafeteria had 23 apples. If they used 20 to make lunch and bought 6 more, how many apples do they have?

====Answer==== Result for Input 0: Roger started with 5 balls. 2 cans of 3 tennis balls each is 6 tennis balls. 5 + 6 = 11. The answer is 11. (confident)

Result for Input 1: Roger started with 5 balls. 2 cans of 3 tennis balls each is 5 tennis balls. 5 + 5 = 10. The answer is 10. (not confident)

Result for Input 2: The cafeteria had 23 apples originally. They used 20 to make lunch. So they had 23 - 20 = 3. They bought 6 more apples, so they have 3 + 6 = 9. The answer is 9. (confident)

| bs/vr/model | | gpt-3.5-turbo | | | GPT-4 | | |
|---|---|---|---|---|---|---|---|
| | | mv | sw-mv | sw-mv-neg | mv | sw-mv | sw-mv-neg |
| 16 | 1 | 76.3 | 73.1 | 63.7 | 87.2 | 87.5 | 86.6 |
| | 3 | 79.4 | 73.4 | 68.1 | 87.5 | 86.6 | 87.8 |
| | 5 | 78.8 | 75.3 | 72.2 | 87.8 | 87.5 | 87.2 |
| | 7 | 78.4 | 75.6 | 71.3 | 87.8 | 87.5 | 88.2 |
| | 9 | 80.0 | 76.3 | 71.9 | 87.8 | 87.5 | 87.2 |
| 32 | 1 | 69.7 | 67.8 | 56.9 | 86.6 | 85.9 | 85.9 |
| | 3 | 74.7 | 68.8 | 55.6 | 87.2 | 86.3 | 87.2 |
| | 5 | 77.5 | 67.8 | 60.6 | 87.5 | 87.2 | 86.3 |
| | 7 | 78.1 | 69.1 | 60.6 | 86.6 | 86.3 | 87.5 |
| | 9 | 76.6 | 68.8 | 60.6 | 87.2 | 85.9 | 86.9 |
| 64 | 1 | | | | 84.7 | 82.5 | 87.5 |
| | 3 | | | | 86.6 | 86.3 | 87.5 |
| | 5 | | | | 85.0 | 87.8 | 87.8 |
| | 7 | | | | 84.7 | 86.6 | 86.6 |
| | 9 | | | | 85.0 | 86.6 | **87.9** |
| 1 | 1 | **80.3** | 77.5 | 79.1 | 87.2 | 86.6 | 86.9 |

Table 8: Comparisons on QQP dataset.

**Boolq:**

Input 0: Passage: Property tax – Property tax or 'house tax' is a local tax on buildings, along with appurtenant land. It is and imposed on the Possessor (not the custodian of property as per 1978, 44th amendment of constitution). It resembles the US-type wealth tax and differs from the excise-type UK rate. The tax power is vested in the states and is delegated to local bodies, specifying the valuation method, rate band, and collection procedures. The tax base is the annual rental value (ARV) or area-based rating. Owner-occupied and other properties not producing rent are assessed on cost and then converted into ARV by applying a percentage of cost, usually four percent. Vacant land is generally exempt. Central government properties are exempt. Instead a 'service charge' is permissible under executive order. Properties of foreign missions also enjoy tax exemption without requiring reciprocity. The tax is usually accompanied by service taxes, e.g., water tax, drainage tax, conservancy (sanitation) tax, lighting tax, all using the same tax base. The rate structure is flat on rural (panchayat) properties, but in the urban (municipal) areas it is mildly progressive with about 80% of assessments falling in the first two brackets.

Question: is house tax and property tax are same

Input 1: Passage: Property tax – Property tax or 'house tax' is a local tax on buildings, along with appurtenant land. It is and imposed on the Possessor (not the custodian of property as per 1978, 44th amendment of constitution). It resembles the US-type wealth tax and differs from the excise-type UK rate. The tax power is vested in the states and is delegated to local bodies, specifying the valuation method, rate band, and collection procedures. The tax base is the annual rental value (ARV) or area-based rating. Owner-occupied and other properties not producing rent are assessed on cost and then converted into ARV by applying a percentage of cost, usually four percent. Vacant land is generally exempt. Central government properties are exempt. Instead a 'service charge' is permissible under executive order. Properties of foreign missions also enjoy tax exemption without requiring reciprocity. The tax is usually accompanied by service taxes, e.g., water tax, drainage tax, conservancy (sanitation) tax, lighting tax, all using the same tax base. The rate structure is flat on rural (panchayat) properties, but in the urban (municipal) areas it is mildly progressive with about 80% of assessments falling in the first two brackets.

Question: is house tax and property tax are same

Input 2: Passage: Pardon – The pardon power of the President extends only to an offense recognizable under federal law. However, the governors of most of the 50 states have the power to grant pardons or reprieves for offenses under state criminal law. In other states, that power is committed to an appointed agency or board, or to a board and the governor in some hybrid arrangement (in some

states the agency is merged with that of the parole board, as in the Oklahoma Pardon and Parole Board).

Question: can the president pardon someone convicted of a state crime

Input 3: Passage: Pardon – The pardon power of the President extends only to an offense recognizable under federal law. However, the governors of most of the 50 states have the power to grant pardons or reprieves for offenses under state criminal law. In other states, that power is committed to an appointed agency or board, or to a board and the governor in some hybrid arrangement (in some states the agency is merged with that of the parole board, as in the Oklahoma Pardon and Parole Board).

Question: can the president pardon someone convicted of a state crime

Input 4: Passage: Jurassic World: Fallen Kingdom – Filming took place from February to July 2017 in the United Kingdom and Hawaii. Produced and distributed by Universal Pictures, Fallen Kingdom premiered in Madrid on May 21, 2018, and was released internationally in early June 2018 and in the United States on June 22, 2018. The film has grossed over $1.2 billion worldwide, making it the third Jurassic film to pass the mark, the third highest-grossing film of 2018 and the 13th highest-grossing film of all time. It received mixed reviews from critics, who praised Pratt's performance, Bayona's direction, the visuals, and the "surprisingly dark moments", although many criticized the screenplay and lack of innovation, with some suggesting the series has run its course. An untitled sequel is set to be released on June 11, 2021, with Trevorrow returning to direct.

Question: will there be a jurassic world fallen kingdom sequel

Input 5: Passage: The Tudors – Showtime announced 13 April 2009, that it had renewed the show for a fourth and final season. The network ordered 10 episodes that were first broadcast on 11 April 2010. The series finale was broadcast on 20 June 2010. The final season was shown in Canada on CBC starting 22 September 2010, and ending on 23 November 2010. Question: is there a season 5 of the tudors

====Answer==== Label for Input 0: [class 1]('confident') Label for Input 1: [class 0]('not confident') Label for Input 2: [class 0]('confident') Label for Input 3: [class 1]('not confident') Label for Input 4: [class 1]('confident') Label for Input 5: [class 0]('confident')

Few-shot examples of QQP and RTE are in the same format of Boolq, therefore we do not provide more examples here.

## H CALCULATION FOR TOKEN

We can calculate the token we need in BatchPrompt. An equation could be as below.

$$l_{total} = l_{task} \times N/s + l_{data} \tag{4}$$

Here $l_{total}$, $l_{task}$, and $l_{d}ata$ are the token we need in total, task specification, and the whole data respectively. N is the total number of data, S is the batch size. Let's take Boolq as an example. For Boolq, the task specification length is 501, the data length is 24011. When batch size is 1, the total token we need is 501*320+24011 = 184331. And this number will be 501*20+24011=34031 for batch size 16, and 501*10+24011=29031 for batch size 32.

We have to emphasize that when batch size is larger than 16, the decrease of token number is not that obvious. However, we have to point out that we can save a large number of LLMs calls. When the batch size doubles, the LLMs calls will decrease by half. As most companies have a limit of quotes to call LLMs, such LLMs call saving is significant.

## I PUTTING THE DATA AT THE VERY BEGINNING OR END

As mentioned in Liu et al. (2023a) and shown in Fig.1, we can find the data putting at the very beginning or end can get better performance. We have to point out two limitations. First, ignoring the LLMs generated results for data in the middle will be waste of token, which will bring much more repeated experiments; Second, the exact range of middle, depending on the data length and

the batch size, cannot be defined. Also, as we find for all the three datasets, the performance of BatchPrompt is already competitive to SinglePrompt, a further boost of accuracy is not easy to achieve only through putting data at specific positions (Beginning, end). Therefore we do not need the repeated Therefore, we still keep our random permutation strategy in the paper.

