# OpenReview forum: "BatchPrompt: Accomplish more with less"
_ICLR.cc/2024/Conference — ICLR 2024 poster_

### Official Review · Reviewer_Nbfh · 2023-10-30

**Soundness:** 3 good
**Presentation:** 3 good
**Contribution:** 3 good
**Rating:** 6
**Confidence:** 4

**Summary:**

This paper proposes an efficient prompting technique, BatchPrompt, which batches the input samples into a single prompt to improve the token utilization. While simply batching samples leads to a significant performance drop, this paper introduces Batch Permutation and Ensembling (BPE) and Self-reflection-guided Early Stopping (SEAS) to maintain the generation quality. BPE permutes the data order in each batch and uses majority voting to get the final prediction. SEAS allows early stopping of voting when LLM is confident about the sample. Experiments on some language understanding tasks show BPE+SEAS boosts BatchPrompt performance to be competitive with single-data prompting while using far fewer tokens and API calls.

**Strengths:**

- The idea of BatchPrompt is simple and practical. Using ensemble and early stopping techniques BPE and SEAS to improve performance is novel to me.
- The paper is clearly written and easy to follow.
- The work focuses on the important problem of improving the prompting efficiency of LLM inference.

**Weaknesses:**

- The proposed method adds some hyperparameters like batch size and voting rounds for configuration. More analysis could be provided on computational efficiency tradeoffs and how to determine the good hyperparameters
- The experiments are conducted on language understanding tasks. It would be helpful to evaluate the method on more diverse tasks, e.g. reasoning, knowledge-intensive QA, and creative writing.

**Questions:**

- How to determine good hyperparameters like batch size and voting rounds for BatchPrompt? More analysis could be provided on computational efficiency tradeoffs.
- It seems that gpt-3.5-turbo suffers from performance degradation when using BatchPrompt, while gpt-4 does not. Is this caused by the model scale? I believe it is helpful to add an analysis of BatchPrompt on the LLaMA series with different model sizes.
- It would be helpful evaluate the method on more diversed tasks, e.g. reasoning, knowledge-intensive, creative writing tasks. Does the type or difficulty of the instruction affect the performance of BatchPrompt?

---

> ### Author Response · Authors · 2023-11-16
> **Response to reviewer Nbfh**
>
> ### [Q1]
> Thank you. First, we set the batch size 32 and voting rounds 5 as default. We provide the detailed token usage against voting rounds/batch size in Fig. 4, as well as in our “general response to reviewer PJJX, pc53, Nbfh”, with detailed numbers. Detailed explanation for choosing batch size and voting round in our response to R3-Q2.
>
> In general, hyperparameter choices can be important. The ablation on batch size and voting rounds are intended to illustrate this. While there is no absolute strategy guaranteed to work on all tasks or in all domains, “batch size 32 and voting rounds 5” is a good default setting, and a short calibration is possible for any task. This process is similar to fine-tuning the parameter of neural networks. We cannot assume the best batch size/epoches when training a neural network at the very begining, but need a validation process to achieve this. To find the best parameters for a specific task, we can use a small validation dataset with ~100 randomly chosen data to validate. For example, a user can start from batch size 16 and 1 voting rounds, and iteratively increase batch size or voting rounds or both, until performance stabilizes, and according to priority of cost-savings and accuracy. The optimal parameters still depend on users’ needs and priorities.
>
> ### [Q2]
> Thanks to the reviewer for the kind suggestion. GPT-3.5-turbo indeed suffers more from performance degradation compared with gpt-4, and it is caused by the model scale. BatchPrompt performance improves as LLM performance improves.. Also, it has been the trend that more advanced LLMs allow more tokens (32k for Gpt-4, 8k for GPT-3.5-Turbo), which also makes larger batches possible. In this way, every update that increases the available context window of LLMs adds to the leverage now possible with BatchPrompt.
>
> Benchmarking with Llama is a good idea, and will make a good addition to our revision. We agree with your hunch, that a larger more generally-performant model will improve BatchPrompt performance.
>
> ### [Q3]
> We also evaluate the method in GSM8K, which is an arithmetic reasoning task. Also, the task presented in  our “general response to reviewer PJJX, pc53, Nbfh”.
> BatchPrompt should generally be applicable to all close-ended tasks, e.g., reasoning, translation, summarization, etc., while might be difficult to be applied to tasks with open-ended solutions/answers, e.g., text generation, creative writing, etc.
> On difficulty of instruction, we agree that a difficult-to-answer, or ambiguously-worded instruction could hamper the performance. Finding the best instruction should be important, while might not be the major task for BatchPrompt. We will leave this to future work.

---

### Official Review · Reviewer_pc53 · 2023-10-30

**Soundness:** 4 excellent
**Presentation:** 3 good
**Contribution:** 3 good
**Rating:** 8
**Confidence:** 3

**Summary:**

The paper proposes a method for batching prompts.  Larger batch size generally improve throughput, but degrade performance.  This paper introduced some suggestions (voting rounds and SEAS) to reduce the performance degradation.

**Strengths:**

The paper advocates the use of batching for prompting, and may be successful in setting a new trend in that direction.

**Weaknesses:**

I worry about running so many experiments.  The plots in Figure 3 suggest that there are patterns to the results, but even so, if we run lots and lots of experiments and report the best values, the best value could be the result of randomness.

On the other hand, to make the case for trends, we may need to run even more experiments over more benchmarks, models, batch sizes and so on.

It would be nice to fit some kind of smooth regression to the results to help with interpretation.  Can you say how performance depends on batch size, voting rounds and model?  An ANOVA would help address concerns above with running so many experiments.

**Questions:**

Can you say more clearly up front that large batches improve throughput, but would degrade performance.  To address performance, you introduce voting rounds and SEAS.  This should also be stated clearly in the conclusions.

The discussion of the results should address the comments above about interpretation.  The ablation studies show that voting rounds are effective.  But it is hard to see the relation between batch size and performance.  It looks like batch size still reduces performance, even with voting rounds and SEAS.  Is that right?

---

> ### Author Response · Authors · 2023-11-17
> **Response to Reviewer pc53**
>
> ### [Q1]
> Good suggestion. We experimentally find that large batches improve throughput while degrading performance, which can be mitigated through BPE+SEAS. We will clarify this in the intro and conclusion in the next version.
>
> ### [Q2]
> Thank you for these comments.
> “The ablation studies show that voting rounds are effective. But it is hard to see the relation between batch size and performance.”
> “Can you say how performance depends on batch size, voting rounds and model? An ANOVA would help address concerns above with running so many experiments.”
>
> In general, we see that performance improves as number of voting rounds increases - this is natural, as we expect that with more samples for each data point, we converge on the true result (marginalizing out the effect of batch position). We understand that [on their own] larger batch sizes tend to decrease performance — an effect that is strongly mitigated by multiple voting rounds. Therefore, the optimal batch size is that which achieves the efficiency target, paired with a number of voting rounds sufficient to keep performance above a threshold. We empirically found that batch size 32 was the best choice across tasks. Equivalently, a practitioner could set the number of voting rounds to 5 by default (after which we empirically saw diminishing returns), and continue boosting batch size as long as performance remains high.
>
> For more details on degradation due to large batch size, we refer to [2307.03172] Lost in the Middle: How Language Models Use Long Contexts (arxiv.org). Due especially to this effect, we find that BPE is an important component to building efficient systems that still perform well.

---

### Official Review · Reviewer_7rFs · 2023-11-03

**Soundness:** 3 good
**Presentation:** 2 fair
**Contribution:** 2 fair
**Rating:** 5
**Confidence:** 4

**Summary:**

For NLP tasks where each data point for inference is not necessarily lengthy, the token count for instructions and few-shot examples in the prompt may be considerably larger than that of the data point, resulting in lower token-resource utilization. This paper try to alleviate the preceding problem by batching multiple data points into a single prompt, a prompting strategy we refer to as “BatchPrompt”. This strategy increases the “density” of data points, which in turn leads to improved token utilization, which shows promising fulture.

**Strengths:**

1.	Batchprompt could highly improve token-resource utilization
2.	BPE could effectively It can effectively reduce the error rate caused by the different position in a batch.
3.	SEAS could effectively reduce the amount of unnecessary calculations

**Weaknesses:**

1.	It seems that each item in the new batch (with only one prompt) could not be computed parallelly as original. Whether it will increase the time cost? It might be better to add time and flops metrics in the experiments.
2.	I think the “batchprompt” could be used in both training and test phases, right?
3.	In BPE, the weight for confidence is directly 1. What about to generate the weights scores directly by the LLM without whether confident?

**Questions:**

Please see the weakness.

---

> ### Author Response · Authors · 2023-11-15
> **Response to reviewer 7rFs**
>
> ## [Q1]
> Thank you for the question. As our motivation is frugality for LLMs, we initially did notconsider the time cost. We use OpenAI API, and inference time for GPT can depend on number of concurrent users, which is not stable. A statistic can be found below: https://gptforwork.com/tools/openai-api-and-other-llm-apis-response-time-tracker . A detailed comparison of response time against tokens:
>
> https://community.openai.com/t/gpt-3-5-and-gpt-4-api-response-time-measurements-fyi/237394 . Also, as we are directly using pre-trained LLMs (gpt-3.5-turbo, gpt-4), we might not be able to calculate an accurate number of flops.
>
> In the link above, we see that response time is in direct proportion to number of completion tokens. Since we use many fewer tokens compared with SinglePrompt (see token comparison in Table 2-4, Fig. 4), we will use proportionally less time.
>
> ## [Q2]
> Very interesting idea. We did not consider BatchPrompt for training efficiency, only for robust inference. This could be a nice area of future research.
>
> ## [Q3]
> Right. Weights can be generated by the LLM. We simplify to 1 and \alpha, for sake of exposition. We can also generate a score directly. Please refer to Appendix D, where we have a description:
>
> ```
> [Conf-Description]: ’You not only need to generate The label/answer, but also your confidence. If you are confident in your output class, append a ”(confident)” at the end of the label; else, append a ”(not confident)”.’
> ```
>
> Here, we could  change this description to:
>
> ```
> [Conf-Description]: ’You not only need to generate The label/answer, but also your confidence. Please append a confidence score between 0 and 1 at the end of the label to represent your confidence in the result you generate.
> ```
>
> Then we can multiply the score with the corresponding result, and select the result with maximum score as final output in the majority voting phase.
>
> We experimented with allowing a range of confidence values, but the results did notchange. Therefore we chose the binary representation. We will explain this in the next paper version.

---

### Official Review · Reviewer_PJJX · 2023-11-03

**Soundness:** 3 good
**Presentation:** 3 good
**Contribution:** 3 good
**Rating:** 6
**Confidence:** 4

**Summary:**

The paper describes a method to improve resource utilization by increasing the 'density' of user query tokens using batching the queries. User queries exhibit lower token utilization compared to the system prompts and/or few shot examples that goes with the query. Authors point out that this is not cost efficient and the 'batchprompt' method requires less LLM calls and better user query token utilization (saving the overall numbers of tokens in a batch which in effect is more cost-efficient).

Batch prompting makes the LLM generation task n times harder for batch size n since the LLM needs to generate n outputs corresponding the n packed queries. Authors conduct experiments and show that this significantly degrades performance, and the order of the packed queries also significantly impact the performance.

Authors develop a batch permutation and ensembling method (to utilize voting from repeated permutations) - this slightly increases the token count and increases the LLM calls (still much less compared to single prompt inference) however improves the performance. Further improvement is realized with 'self reflection guided early stopping (SEAS) scheme) where the generator is also asked to provide the confidence of the result and using rules, the system stops the voting procedure early.

Authors have performed experiments on 3 datasets (yes/no question answering, entailment, paraphrase detection) and shown that with a batch size of 32, and using BPE and SEAS. the accuracies on 3 datasets do not degrade (improve slightly).

**Strengths:**

- Authors propose a robust method that uses larger batch size, more voting rounds (eg. 5+) and a self-reflection guided early stopping approach.

- The early stopping method also uses a pruning strategy to prune away confident predictions leaving fewer/harder samples for later
rounds. In the process, the harder samples might also become easier to predict, due to smaller effective batch size in later rounds.

- via experiments, authors show that voting is most successful when the baseline performance is strong (for example, gpt4 vs. gpt3.5)

**Weaknesses:**

- Authors chose small number of tasks (only 3 simple tasks (yes/no QnA, paraphrase detection and entailment detection) -> these tasks may be too easy for gpt3.5 and gpt4 systems

- Results are shown using few experiments (~300 dataset queries each for the 3 datasets); typically a validation on more tasks and more datasets would have helped get a more confident understanding of the approach.

- this is a nice applied research paper with good results and a principled approach for improving cost efficiency, however there are many variables to unpack (quality and length of tasks, mixing different types of instructions, performance on novel datasets not seen by the LLMs, solving position bias discrepancy via BPE with more experiments and results, role of prompt variations on the results, etc)

**Questions:**

- All tasks are very short answer type tasks, using tasks that generate longer answers might be very hard to experiment using the batchprompt approach. Couldn't see a discussion on this topic in the paper. Thoughts?

- it is not clear how this system would be used with all its advantages in a deployment scenario - batching real world prompts with very different instructions might have unpredictable behavior, any thoughts?

---

> ### Author Response · Authors · 2023-11-15
> **Response to reviewer PJJX**
>
> ## [Q1]
> Sorry for the lack of clarity. BatchPrompt can be used for all close-ended tasks. Task answers need not be short, but must have the same format in each voting round, in order to determine agreement. This is required for our ensemble (majority voting) step. We achieve this goal through task specification.
>
> Please note the prompt in Appendix D. For all three tasks in the paper, we write the following description at the end our task specification:
>
> ```
> Below are the outputs you need to generate. ”X” can be ’0’ or ’1’.
> Label for Input 0: [class X] [Place-Holder-Conf]
> Label for Input 1: [class X] [Place-Holder-Conf] ......
>
> Please make sure each generated label is in format of [class X].
> Please make sure to generate [BATCH-SIZE] labels. You may include other additional sections here.
> ```
>
> The motivation for writing this is to encourage the model to map answers of varied formats  to a binary label, and to unify the format of the answer to make majority voting possible.
>
> However, what if the answer cannot be binary label?
>
> Please note the work in Appendix C. We provide a general experimental result on GSM8K, the arithmetic reasoning task, whose answers cannot be a binary label. The answer to the arithmetic reasoning task can be long chain-of-thought answers. That said, we can still use the task specification to map the answer to a specific number/format, enabling a majority vote. In Appendix D, we provide prompts for GSM8K:
>
> ```
> Please make sure to write your a series of intermediate reasoning steps. Please make sure the final sentence is ”The answer is xxx.”, and the answer should be a number. Please make sure to generate [BATCH-SIZE] labels each time.
> ```
>
> We add this at the end of each task specification to make answer formats consistent in each voting round, and make ensembling possible.
>
> We recognize that for open-ended tasks like text generation, it is not obvious how BatchPrompt could be applied. BatchPrompt requires an “equality” function to enable majority voting. Suppose two open-ended responses are considered to “say the same thing”. By providing this kind of semantic equality function, BatchPrompt can be extended to this task. We leave this as future work.
>
> ## [Q2]
> If we understand correctly, the question is about batching different tasks into each model call, e.g.
>
> ```
> “Instruction #1, Question #1, Instruction #2, Question #2, …”
> ```
>
> versus
>
> ```
> “Instruction #1, Question A, Question B, Question C, …”
> ```
>
> In the former case, we lose the token-saving benefit from sharing instructions, but still get other robustness and cost-saving benefits from permutation, ensembling, and early stopping. BatchPrompt is designed for processing large volumes of data where the LLM performs a specific function, e.g. “label sentiment of all emails in an organization”, or “map each request to one of k functions/tools/teams”. We expect LLMs will take many such roles in software systems.
>
> ## [Answer to ~300 dataset queries each for the 3 datasets]
> Thank you for bringing this up - we will add clarification.. Each benchmark dataset is already divided into train/validation/test by Hugging Face, and we directly use the validation data in our experiments. We do not, however, use the full validation set for two reasons: (1) Quota constrains our ability to test, and (2) Data containing sensitive content cannot be used with gpt-3.5-turbo/gpt-4.  Therefore, we use gpt-3.5-turbo to filter out sensitive content, and randomly choose 320 samples for Boolq, QQP, RTE.
>
> ## [Answer to small number of tasks]
> One more task, on arithmetic reasoning, can be found in Appendix D. We provide three representative tasks in the main paper, but emphasize that BatchPrompt can be used in any task with close-ended answer, as mentioned in the answer to Q1.
>
> To increase confidence, we will add the following results on more datasets, with long answers, and where all validation (>300) data is used. This can be found above, as in our "general response to reviewer PJJX, pc53, Nbfh".
>
> ## [Answer to these tasks may be too easy]
> The major motivation for BatchPromptis frugality, while maintaining base-level accuracy (i.e. “SinglePrompt”), for tasks of any difficulty. We do not seek to improve the accuracy of each task, but keep performance stable, while saving energy and money via fewer tokens and LLM calls.  That said, permutation and ensembling might be even more useful in cases of hard tasks, where repeated samples lower the risk of picking an incorrect answer. This related topic has been studied in: [2203.11171] Self-Consistency Improves Chain of Thought Reasoning in Language Models (arxiv.org)
>
> ## [Answer to many variables to unpack]
> Thank you for the detailed comment and feedback. We indeed have many  parameters/variables, but one easy way to use BatchPromptis to simply use BatchPrompt+BPE+SEAS with default parameters, e.g., set batch size to 32, maximum voting time to 7, and LLMs temperature to 0.

---

> > ### Comment · Reviewer_PJJX · 2023-12-04
> > **thank you**
> >
> > thanks for answering the questions and doing more experiments on COPA and MNLI with good results. I've increased my score slightly based on the responses.

---

### Author Response · Authors · 2023-11-15
**A general response to reviewer PJJX, pc53, Nbfh, with more experimental results**

Thanks for the comments about the task selection problem. Although we only present three tasks in the main paper, BatchPrompt can be generally used for all close-ended tasks.

One more task, on arithmetic reasoning, can be found in Appendix C. We provide three representative tasks in the main paper, but emphasize that BatchPrompt can be used in any task with close-ended answer, as mentioned in the answer to Q1.

To increase confidence, we will include the following raw results on additional datasets, that have long answers, and where all validation data (>300) is used.:


## Results on Copa(Choice of Plausible Alternatives):
### ChatGPT
| Accuracy | VT=1   | VT=3  | VT=5  | VT=7  | VT=9  |
|---       |---     |---    |---    |---    |---    |
| BS=1     |89.0625 |   -   |   -   |   -   |   -   |
| BS=16    |82.8125 |82.8125|85.5   |85.9375|85.9375|
| BS=32    |70.3125 |76.5625|79.6875|85.9375|85.9375|




| Token Num | VT=1  | VT=3  | VT=5  | VT=7  | VT=9  |
|---        |---    |---    |---    |---    |---    |
| BS=1      |37466  |   -   |   -   |   -   |   -   |
| BS=16     |4646   |9642   |9833  |10112  |10112  |
| BS=32     |3552   |7961  |8324  |8525  |8525  |




| Calling Num | VT=1    | VT=3  | VT=5  | VT=7  | VT=9  |
|---          |---      |---    |---    |---    |---    |
| BS=1        |64       |   -   |   -   |   -   |   -   |
| BS=16       |4        |12     |20     |28     |36     |
| BS=32       |2        |6      |10     |14     |18     |




### GPT4
| Accuracy | VT=1   | VT=3  | VT=5  | VT=7  | VT=9  |
|---       |---     |---    |---    |---    |---    |
| BS=1     |96.875  |   -   |   -   |   -   |   -   |
| BS=16    |98.4375 |98.4375|98.4375|98.4375|98.4375|
| BS=32    |98.4375 |98.4375|98.4375|98.4375|98.4375|


| Token Num | VT=1  | VT=3  | VT=5  | VT=7  | VT=9  |
|---        |---    |---    |---    |---    |---    |
| BS=1      |37466  |   -   |   -   |   -   |   -   |
| BS=16     |4646   |9862   |9862   |9862   |9862   |
| BS=32     |3552   |8468   |8468   |8468   |8468   |



| Calling Num | VT=1 | VT=3 | VT=5  | VT=7  | VT=9  |
|---          |---   |---   |---    |---    |---    |
| BS=1        |64    |   -  |   -   |   -   |   -   |
| BS=16       |4     |12    |20     |28     |36     |
| BS=32       |2     |6     |10     |14     |18     |

## Results on MNLI (Natural Language Inference on MultiNLI)
### ChatGPT
| Accuracy | VT=1   | VT=3  | VT=5  | VT=7  | VT=9  |
|---         |---       |---    |---      |---    |---      |
| BS=1     | 77.5%  |   -   |   -   |   -   |   -   |
| BS=4     | 66.3%  | 68.2% | 72.1% | 72.7% | 72.7%   |
| BS=16    | 64.2%  | 63.4% | 71.2% | 71.6% | 72.3% |



| Token Num | VT=1  | VT=3  | VT=5  | VT=7  | VT=9  |
|---          |---    |---      |---      |---    |---      |
| BS=1      |158401 |   -   |   -   |   -   |   -   |
| BS=4      | 51361 | 82723 | 90085| 90447| 90447|
| BS=16     | 24601 | 55963 | 62325 | 64435 | 65524 |

| Calling Num | VT=1    | VT=3  | VT=5  | VT=7  | VT=9  |
|---            |---      |---      |---      |---    |---      |
| BS=1        | 320     |   -   |   -   |   -   |   -   |
| BS=4        | 80      |  240  |  400  |  560  |  720  |
| BS=16       | 20      |  60   |  100  |  140  |  180  |

### GPT4
| Accuracy | VT=1   | VT=3  | VT=5  | VT=7  | VT=9  |
|---         |---       |---    |---      |---    |---      |
| BS=1     |  88.8% |   -   |   -   |   -   |   -   |
| BS=64    |  75.3% | 80.3% | 82.8% | 82.2% | 83.1% |


| Token Num | VT=1  | VT=3  | VT=5  | VT=7  | VT=9  |
|---          |---    |---      |---      |---    |---      |
| BS=1      |152321 |   -   |   -   |   -   |   -   |
| BS=64     | 17816 | 49178 | 54540 | 54582 | 54668 |


| Calling Num | VT=1    | VT=3  | VT=5  | VT=7  | VT=9  |
|---            |---      |---      |---      |---    |---      |
| BS=1        | 320     |   -   |   -   |   -   |   -   |
| BS=64       | 5       |  15   |  25   |  35   |  45     |

(as MNLI dataset is large and our GPT-4 quota is quite limited, we only use the maximum batch size (64) as an example to conduct experiments and show the results.)

More results on other datasets, e.g., cola, wic etc., are also available, but due to space limitation, we do not mention them here.

---

### Meta-Review · Area_Chair_6ifM · 2023-12-05

**Metareview:**

This paper proposes a simple yet novel approach to improving the prompting efficiency of LLM inference based on Batch Permutation and Ensembling (BPE), a majority vote approach that improves labelling quality through repeatedly permutating data positions in a batch at the price of more token usage, and Self-reflection-guided EArly Stopping (SEAS) to counterbalance the additional token usage caused by the voting process. In the words of one of the reviewers, the authors present “good results and a principled approach” towards demonstrating the effectiveness of their method. A key weaknesses is the need for more rigorous evaluations on more diverse sets of tasks; however, the authors have presented additional results on other datasets as part of the rebuttal. The authors should ensure that they address all of the reviewers’ comments (and e.g., not only add raw results but thoroughly describe all new results on additional datasets) and run additional experiments as needed (e.g., ANOVA to facilitate interpretation). The authors should also ensure they thoroughly proofread their paper.

Other comments:
It is not clear what the authors mean by “Data containing sensitive content cannot be used with gpt-3.5” — please clarify in the paper. Also, they might want to consider a different abbreviation for their first method as ‘BPE’ is already used in the literature (Byte-Pair Encoding).

**Justification For Why Not Higher Score:**

The nature of the experiments and type of novelty, which is more incremental than truly groundbreaking, would be better presented as a poster.

**Justification For Why Not Lower Score:**

The paper has an average score of 6.25. I do not see any substantial weaknesses to the approach and there is good novelty, and so I believe the paper qualifies as an accept.

---

### Decision · Program_Chairs · 2024-01-16

Accept (poster)